# Stability of Treatment from Earth-Based Mortar in Conservation of Stone Structures in Tanais, Russia

**Ewa Sobczyńska \*, Wojciech Terlikowski and Martyna Gregoriou-Szczepaniak**

The Institute of Building Engineering, Warsaw University of Technology, 00-637 Warsaw, Poland; wterl@il.pw.edu.pl (W.T.); mgreg@il.pw.edu.pl (M.G.-S.)

\* Correspondence: e.sobczynska@il.pw.edu.pl

**Abstract:** Identification of materials, masonry elements, their shapes, physical and mechanical parameters and type of connection is crucial for the conservation works related to ancient masonry structures. In the case of the archaeological site where the research was carried out (Tanais in Russia), some irregular masonries made of limestone and earth-based mortar were stated. Such type of structures is a common finding during the archaeological excavations in the Black Sea basin carried out by the Division of Fundamental of Building of the Civil Engineering Faculty of the Warsaw University of Technology in cooperation with the Institute of Archaeology of the University of Warsaw and the Antiquity in Southeastern Europe Research Center. The structure of such walls is degraded to a large extent, has low strength, internal cohesion and, as a result—low durability. At the same time, due to their historical importance, proper conservation, as well as the development of the whole methodology for selecting the best composition of earth-based mortar, is of great importance. Presented in the article, research on earth-based mortars were carried out to determine the best way to strengthen them, using cement (creating an earth-based mortar stabilized with cement with the most appropriate recipe) and other substances available in the region where conservation works are carried out not only to improve the durability physical and mechanical parameters but also to achieve the desired esthetic effect in the form of a suitable tone together with the compatibility of repair mortar with the substrate and constitutes the primary stage of creating the whole methodology of selecting a proper composition of earth-based mortar for the conservation of ancient stone structures. In this stage, four criteria were taken into consideration: mechanical (compressive strength test), conservation (compatibility, reversibility, color, texture and surface profile), durability (freeze–thaw test, the appropriate finish of the surface, shrinkage, workability) and technological one (application of materials, technology and techniques available at the conservation area). Applied treatment was evaluated in the next two years of the conservation works. Parameters of repair earth-based mortar stabilized with cement fulfilled all of the above-mentioned requirements.

**Keywords:** conservation; earth-based mortar; ancient stone structures; masonry

## 1. Introduction

The basis for the effectiveness of all activities related to maintenance, strengthening, rehabilitation of the historic masonry structures and preparation for their later exposure is the correct diagnosis. Detailed identification of materials, technology and techniques for the construction of the historic building should be a part of it. Building materials, as well as the techniques and technologies used in construction today, are, in principle, well-known. In the case of ancient buildings, we often deal with techniques and technologies not used today, and often archaeological discoveries show techniques that today are known only locally or completely unknown. Proper understanding of the load-bearing structure and its work resulting from the real static scheme is sometimes difficult, but very important.

The proper recognition of the construction technique, the physical and mechanical properties of the materials used give the opportunity to properly assess the technical

condition of the structure or part thereof, correctly determine the work of load-bearing structures, including issues related to the structural rigidity of the historic buildings. Knowledge of the physical and mechanical properties of materials used in the construction of historical buildings gives the possibility of proper verification of structural strength, as well as the adoption of an appropriate rehabilitation solution, security measures and strengthening. All materials used in the conservation activities should be original or compatible with the existing masonry so as to not cause damages or be detrimental to the existing fabric and be durable as possible under those requirements [1].

The conservation aspect shows the relationship of the materials, techniques and technologies originally used with activities aimed at securing and preserving the monument, stopping its destruction and documentation of these activities. These activities are conditioned by international arrangements and principles, including the principle of compliance of materials, techniques and construction technologies and the principle of reversibility of methods and materials—all conservation activities should be carried out in such a way and using such materials that can be removed in the future, restoring the original state [2].

In the case of the conservation activities in the Tanais, various limestone rocks, mostly unsorted, and earth-based mortar, were used. Such type of binder is nowadays increasingly commonly used not only in the form of masonry and plaster mortars [3–6] but also rammed earth walls [7–9] as well as earth blocks and cobs [10]. We can also find numerous standards dealing with the issues of earth-based materials [11], but differing from each other depending on the country, as well as the basic test results of earth-based mortars [3–5,12]. One crucial aspect is also the stabilization of earth-based material, which can be defined as controlled modification of its structure in order to achieve desired mechanical properties and proper durability [13]. There is a wide range of literature on the stabilization of earth-based materials [14–20]. The most popular stabilizers are lime, cement and sometimes bituminous emulsions [21]. Some synthetic and natural binders, including polymers, resins and adhesives, as well as fiber reinforcement, may also be used. The choice of the stabilizer should depend on the grain size of the mixture and its plasticity index [13] and in the case of the archeological excavations on the accessibility of the material. In the revitalization of historical structures, it is also important to base on standards and state-of-the-art reports on repair mortars for the historic masonry [1,22] and compatibility of the material used with the existing substrate [23–26]. Any literature studies, however, bring results in the development of the problem discussed in this article, which makes the following research the first attempt to solve it. It is also worth mentioning that this topic is important in the case of the development of the sustainability concept. Mortars for conservation produced with raw materials and techniques contribute to sustainability, so it is crucial to create a whole methodology to select the best composition.

## 2. Characteristics of Irregular Stone Masonry in Tanais

### 2.1. Historical Background

The ancient town of Tanais was formerly located probably at the shore of the Azov Sea and at the same time at the mouth of the Don River [27,28]. Nowadays, due to hydrological changes of this area [29,30], it is located on the right bank of the Mertvy Donets.

Tanais was founded at the end of the 60s/at the beginning of the 50 s of the 3rd-century BC [31] by Greek settlers and existed until the middle of the 3rd-century AD when it was totally destroyed. During its history, Tanais was destroyed twice more—at the very end of 1st-century BC by Bosporanean King Polemon and in the middle of 2nd century AD. In the second half of the 4th century and in the first half of the 5th century, the territory of the Tanais was again inhabited and then finally abandoned. The chronology was studied by Arsen'eva et al. [32].

Heretofore, conservation works at the Tanais site carried by authors consist of the three seasons in years 2016–2020. The research was carried out as part of the conservation missions by the Division of Fundamental of Building of the Civil Engineering Faculty of the Warsaw University of Technology in cooperation with the Institute of Archaeology

of the University of Warsaw and the Antiquity in South-eastern Europe Research Center. The first season was oriented on detailed documentation with a 3-d scanner. One of the outcomes of that documentation is a three-dimensional model of the trench XXV, which allows performing precise measurements of structures located inside (Figure 1) [33].

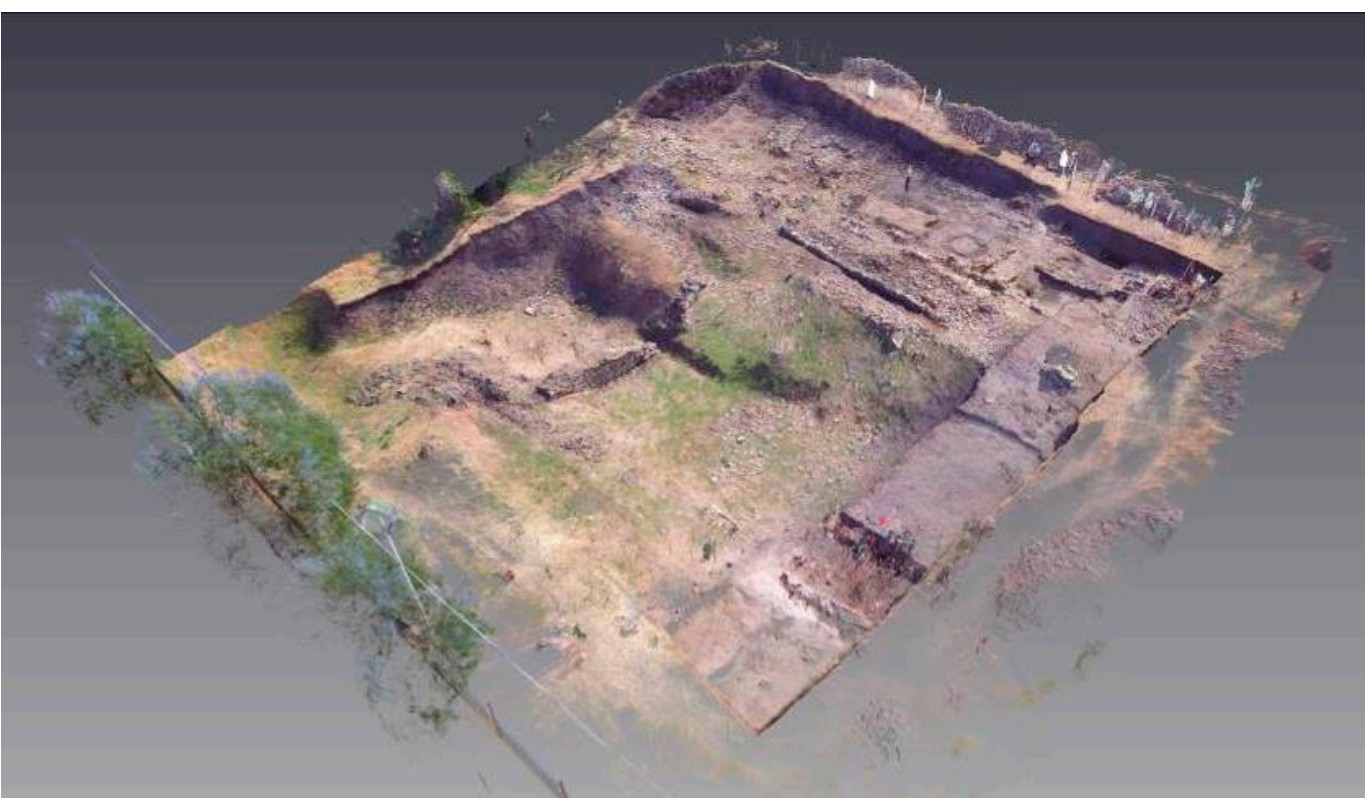

**Figure 1.** Three-dimensional model of the excavation in Tanais, Russia.

These are antique masonry structures made of ashlars and stones from various limestone rocks, shaped or broken, mostly unsorted, of different structure, degree of sedimentation, porosity and absorbability and, what is associated with it, humidity. An earth-based mortar was used as the binder [34,35]. Such a wall has low strength, internal cohesion and, as a result—low durability [36,37]. The structure of the walls is degraded to a large extent, which in addition to possible conscious human impact, results from centuries-long environmental processes, including long-term moisture and biological aggression in the ground and destruction caused by exposure to rainwater. The example of irregular stone masonry before conservation works in the Tanais are shown in Figure 2.

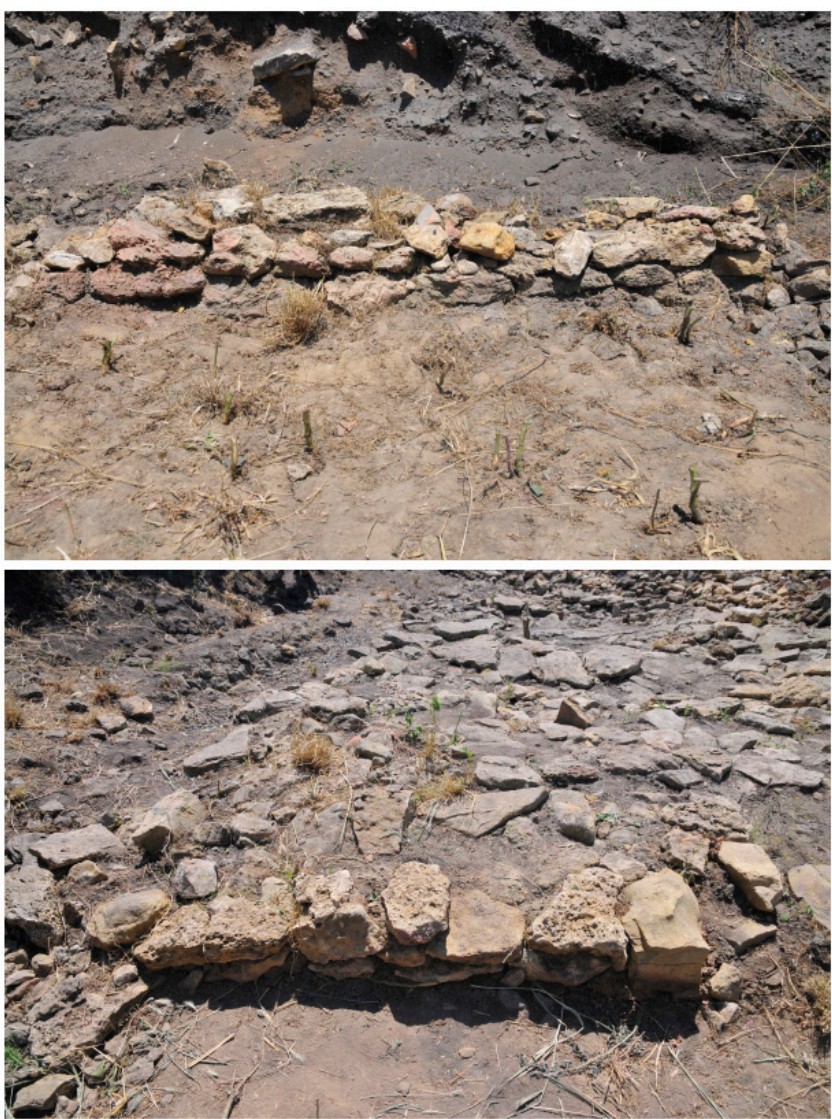

**Figure 2.** Masonry wild stone wall before conservation works, Tanais, Russia.

*2.2. Environmental Conditions*

Environmental impacts are characterized by high temperatures in the summer, with strong solar radiation and low temperatures in winter, with snowfall and negative temperatures causing freezing of water. The literature research confirms that freezing water increase volume—at a temperature of 0 °C it is a 9% increase. That is why the ice appearance is directly followed by internal pressure, which depends on temperature—increases from ca. 0.6 MPa at a temperature of 0 °C, to reach ca. 61 MPa at −5 °C and even 211.5 MPa at −22 °C temperature [38]. As a result of freezing, deformations appear in the masonry structure, which undermines the cohesiveness in the weakest section. Moreover, the displacement and cracks could be observed. The least resistant to this effect are rocks of high water absorption and a finely porous structure. Therefore, the most susceptible to the effects of ice are stones with hairline scratches, crevices and cracks. The water penetrates damages, fills them completely, and freezing bursts, causing deep cracks and disintegration of the wall. The higher frequency of water–ice–water change is, the destruction of masonry is faster. In the case of structures in the Tanais, the additional problem was biological destruction, caused by the damaging effects of green plants, lichens and weeds growing in spring in the trench. There is also heavy rainfall in autumn and spring. These are the impacts that further complicate the conservation works and cause the destruction of the wild walls. The basic problems that need to be solved during the conservation works

are reduction of absorption of water, increasing internal cohesion, related to the need to consolidate the internal wall structure, restoration and protection of damaged parts of walls, reprofiling with the removal of destructive plant interactions.

## 3. Materials and Methods

During the first excavation season, a mortar sample from the existing masonry structure was also collected in order to select the appropriate substitute material for research in Poland to create the best composition of repair mortar for the next excavation seasons. In petrographic studies, a thin plate from the transported material was made—a microscopic slide for examination in polarized light. Initially, this sample was impregnated several times with a solution of xylene and Canadian balsam at a temperature of ~40–50 °C. After the sample had hardened, the thin plate by grinding the sample attached to the glass slide with polishing powders was created in order to obtain a material thickness of about 35–40 μm. After grinding to the target thickness, the thin plate was covered with a coverslip, glued with Canadian balm. It was tested using a Zeiss Axiolab polarizing microscope, with a Canon G2 digital camera mounted for photographic documentation. X-ray diffraction examination of samples was performed with Bruker D8 ADVANCE powder diffractometer, equipped with a copper lamp (current of 36 mA and 36 kV). Powder formulations, non-oriented, ranging from angular measurement 4.0–75.0°2Theta, and oriented (raw, roasted up to 500 °C, and soaked with ethylene glycol) of the clay fraction separated in water suspension was tested. The measurement step was 0.02°2Theta, and the time for all measurements for each step was 1.0 s. The sample was also tested with the thermal method, using differential scanning calorimetry combined with thermogravimetry. A PerkinElmer STA 6000 thermal analyzer was used. Data recording and interpretation were performed with the Pyris software. Sample disintegrated in an agate mortar were analyzed under the following conditions: (1) weight 81.9 mg (2) ceramic crucible, open, (3) temperature range of measurement 40–999 °C, (4) heating rate 15 °C/min (5) $N^2$ atmosphere (flow 20 mL/min).

The sieve and aerometric analysis of a sample of soil from the Tanais was also carried out based on PN-EN ISO 14688 [39] to get the particle size distribution and to name the type of soil. Sieve analysis is used for sand and gravel fractions. Soil is sifted through sieves with specific mesh sizes, as a result of which grains with appropriate diameters remain on the next sieves. The second step is aerometric analysis, when the amount of smaller fractions can be measured by sedimentation.

As a next step, the properties of earth-based mortar made on the base of substitutive material have been tested. Substitutive material was chosen after studies of original earth-based mortar from the Tanais described in paragraph 4. As first comparison sieve and aerometric analysis was performed—cl = 6%, si = 64%, sa = 28% (type of soil—sasiCl). Then the petrographic, X-ray and thermal analysis were carried out, which results are shown in Figures 3–5. Based on the petrographic studies, it can be concluded that the substitutive sample also consists of crumb components of the psammite to aleurite fraction and the material of the pelitic fraction. The aleurite fraction dominates, while the psammite fraction is in a small amount. The main mineral that is part of the crumb components is also quartz. Minerals from the feldspar group, both alkaline and sodium–calcium varieties, are of minor importance. Based on X-ray diffraction examination, it can be stated that the clay fraction is also dominated by illite. Along with illite, the pelitic fraction also includes kaolinite and trace amounts of chlorite. Substitutive material also has a similar course of thermogram.

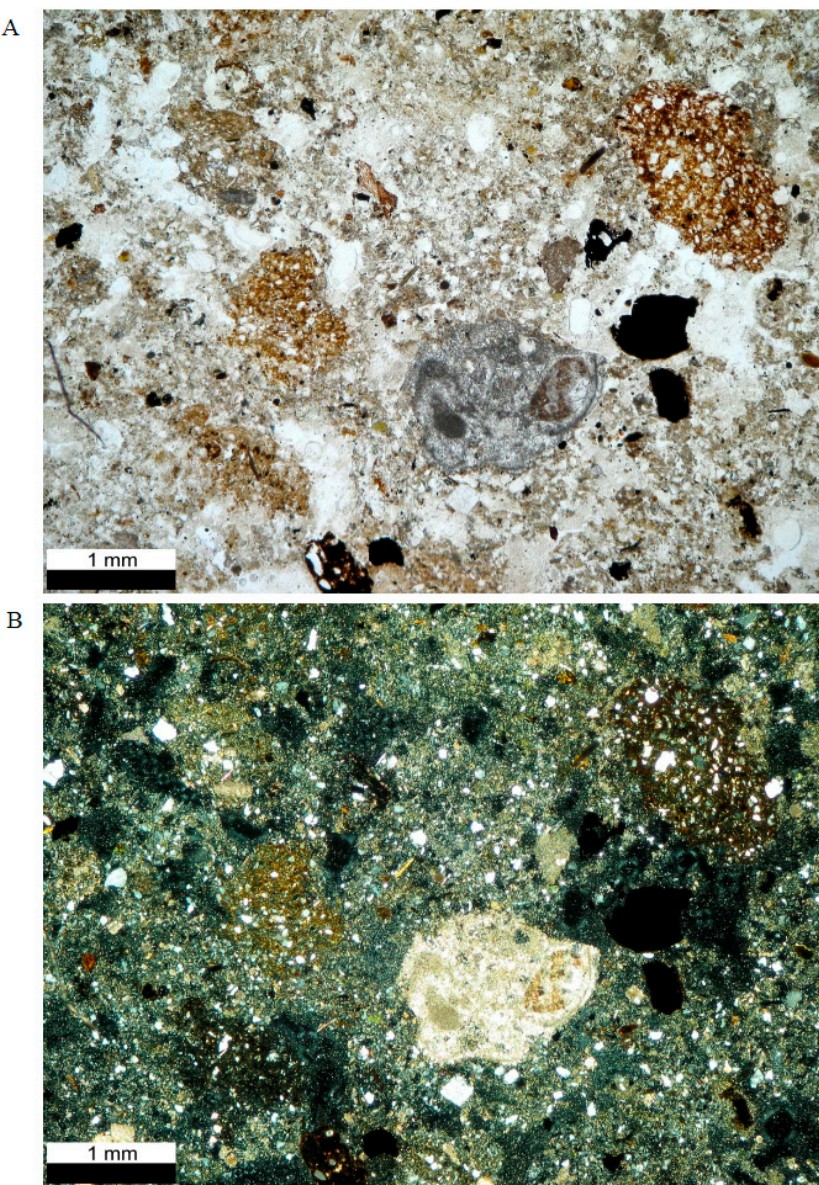

**Figure 3.** Microscopic image seen with one polarizer (**A**) and two crossed polarizers (**B**) [40].

The workability analysis was carried out according to PN-EN 1015–3 [41] using a shock table for the study. A mold was located on the table, where the mortar was placed in two layers. After removing it, 15 shocks were made, and the two diameters perpendicular to each other were measured. The measure of mortar plasticity is the diameter of the mortar cake in centimeters. The result is the arithmetic mean of the two measurements. The compressive strength was tested accordingly to PN-EN 1015–11 [42] and taking into account conclusions from research on the compressive strength testing described in [12]. With regard to the cure, all three samples of dimensions $4 \times 4 \times 16$ cm$^3$ remained in a room of cure for a 28-day minimal period, then were broken into two pieces and tested in compression. Another test was to analyze mortar durability by subjecting it to a freeze–thaw test based on PN-85/B-04500 [43]. Determination of frost resistance is carried out on samples with dimensions $4 \times 4 \times 16$ after 28 days. The frost resistance test consisted of freeze and thaw cycles—one cycle included 4 h in a freezer at $-20$ °C and two hours in water at 20 °C (standard 4 h but after basic test on earth-based mortar when the sample disintegrates just after placing it in water the time was limited to 2 h and additional samples were protected with a hydrophilizing agent—Remmers Funkosil SNL, applied using a well-soaked brush on the whole surface of samples in two layers before the test,

which provide better frost resistance of the material). It was assumed that the weighing of samples would take place after each cycle, and the test will end when a loss of more than 20% of the initial sample weight will be stated. Earth-based mortar samples were also subjected to the test for determination of shrinkage and color analysis. The shrinkage test was performed on samples with dimensions $4 \times 4 \times 16$ after 28 days by comparing the initial sample length with the length after 28 days accordingly to PN-85/B-04500 [43] using a Grauf–Kaufman apparatus. The color analysis was carried out using the basic RAL K7 color palette. Determination of color was a necessary aspect of obtaining a reference level for a repair mortar.

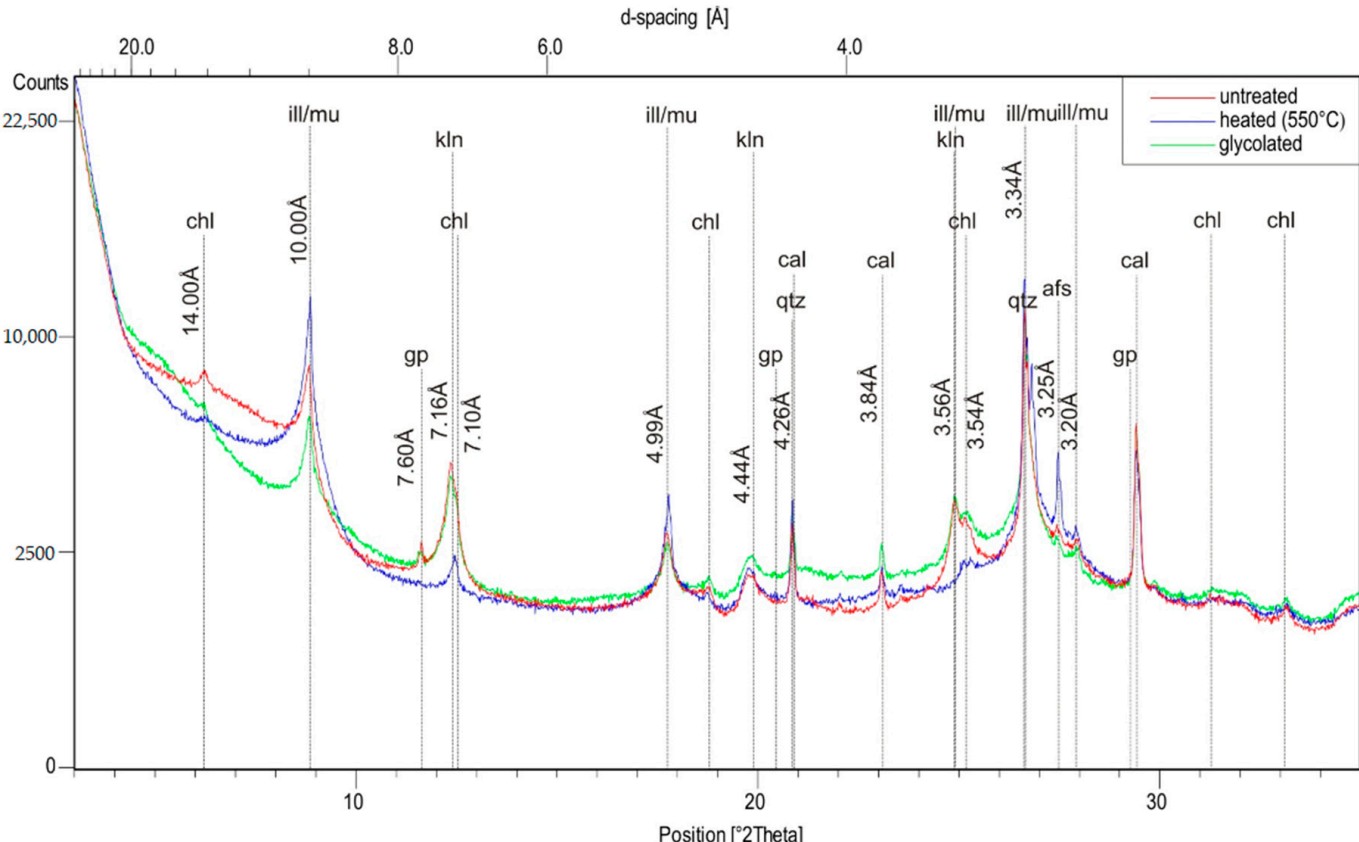

**Figure 4.** X-ray diffraction examination of clay fraction [40] (chl—chlorite, ill/mu—illite/muscovite, kln—kaolinite, qtz—quartz, afs—alkaline feldspar).

In the modified compositions of earth-based mortars, Portland cement CEM I and hydrated lime were additionally used.

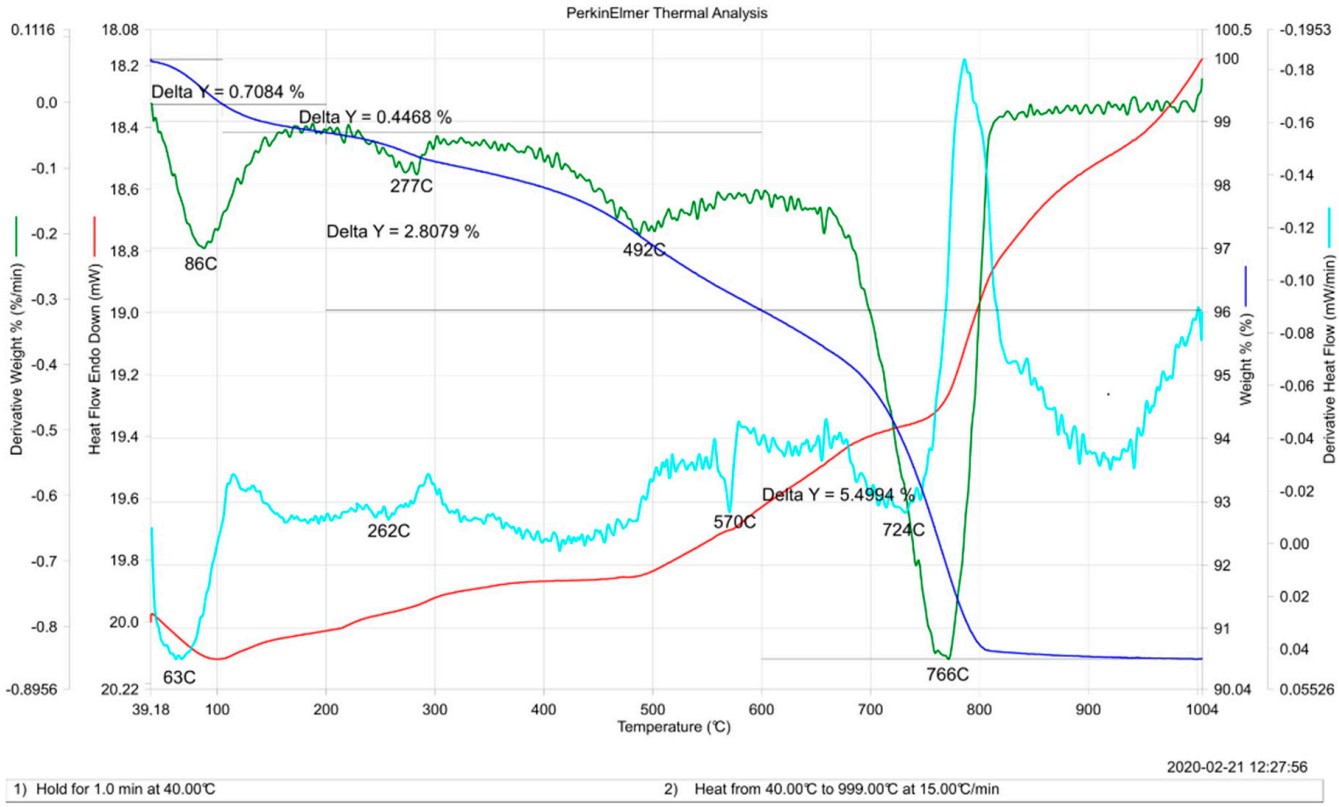

**Figure 5.** Thermogram of the sample [40] (TG—the weight curve-navy blue; DTG—first derivative of the weight curvegreen; DSC—heat flow curve—red; DDSC—first derivative of heat flow curve turquoise).

## 4. Studies on Original Earth-Based Mortar from Tanais

Based on the petrographic studies (Figure 6), it can be concluded that the analyzed original earth-based mortar sample consists of crumb components of the psammite to aleurite fraction and the material of the pelitic fraction. The aleurite fraction dominates, while the psammite fraction is in a small amount. The main mineral that is part of the crumb components is quartz. Minerals from the feldspar group, both alkaline and sodium–calcium varieties, are of minor importance. The grains of rocks were also subordinate and represented the fragments of light crystalline rocks (granitoids, fragments of gneisses, etc.). Additionally, there were fragments of carbonate rocks (limestones, etc.), macroscopically visible as white-gray grains. Such grains mainly consist of calcite. The grains of rocks observed in the composition represent the psammite fraction. A significant amount was also observed within this fraction of quartz grains and black and black–brown grains, which are clusters of opaque minerals (sulfides, oxides, oxyhydroxides) and/or being organic components. The composition of the grain skeleton is complemented by mineralogically varied and at the same time accessory components, mainly representing the so-called heavy minerals, as well as glauconite or mica. The pelitic fraction is definitely dominated by clay components. It is composed of rhombohedral carbonates dispersed within it. Furthermore, within the clay mass, similarly to carbonates, there are dispersed non-crystalline or weakly crystalline iron minerals in the form of oxyhydroxides (goethite, lepidocrocite). A small degree of crystallinity and a lower amount in relation to other components means that they have not been identified by diffraction methods; however, their presence is confirmed by thermal methods. They cause the clay mass to be yellow-orange in color.

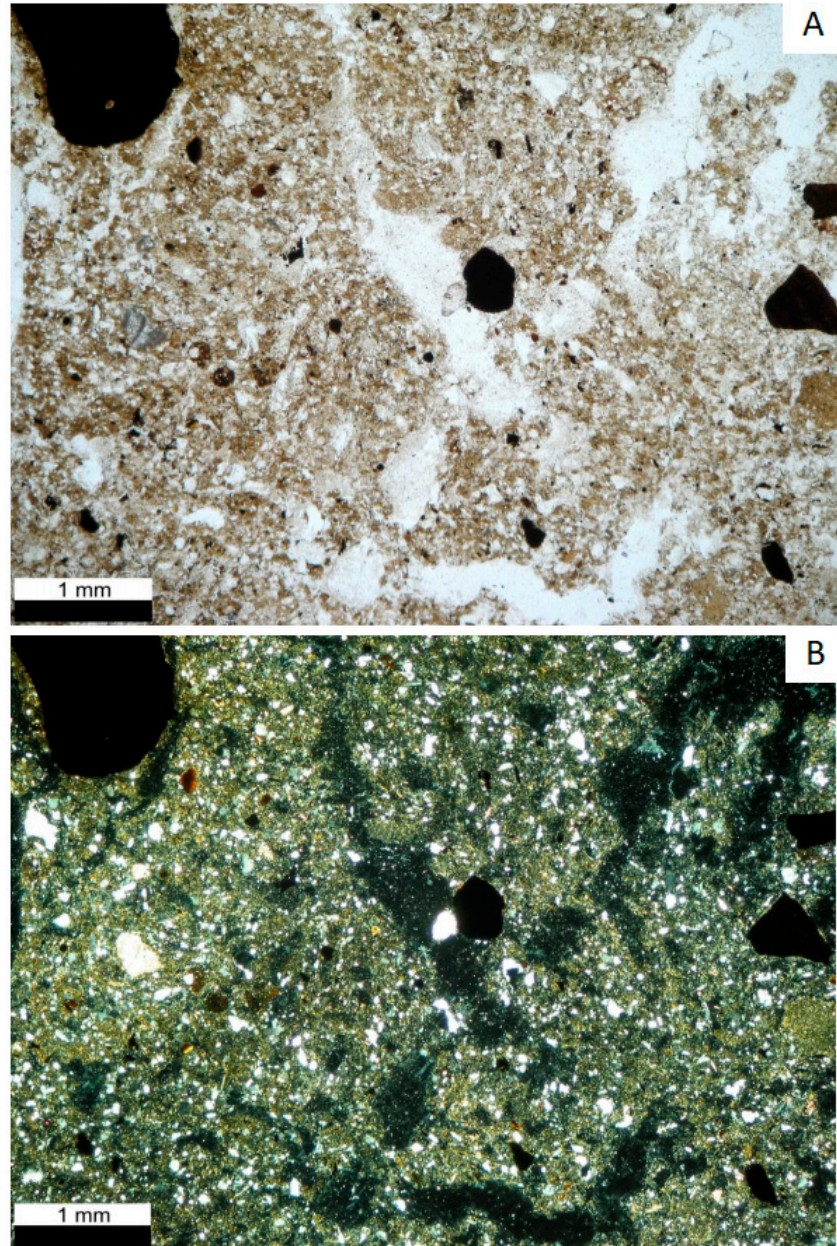

**Figure 6.** Microscopic images are seen with one polarizer (**A**) and two crossed polarizers (**B**) [40].

These mineral phases can also build the above-mentioned black or large black and brown grains of the psammite fraction. Based on X-ray diffraction examination (Figure 7), we can state that the clay fraction has a mixed composition, although it is dominated by illite. Along with illite, the pelitic fraction also includes kaolinite and trace amounts of chlorite. The thermogram of the tested sample is shown in Figure 8. The sample is characterized by low-temperature weight loss, visible as a deflection of the DTG curve (derivative weight curve—green one), and a corresponding endothermic effect (derivative heat flow curve (DDSC)—turquoise one), visible up to 200 °C. In the initial phase (up to 100 °C), these effects should be interpreted as the release of free (hygroscopic) water by the sample. The maximum of this process is visible in the form of the inflection point of the DTG curve, which is below 100 °C. However, the DTG curve, although it returns to the baseline at about 200 °C, after exceeding 100 °C, still indicates a (decreasing) weight loss until reaching the subhorizontal course. Such behavior should be interpreted as the effect of thermal decomposition (dehydration) of clay minerals—illite/muscovite. In the range of 200–400 °C, clear weight and endothermic effects are visible. Maximum weight

loss occurs at 280 °C and 277 °C. It is caused by the decomposition of iron oxides in the form of goethite (α-FeOOH) and lepidocrocite (γ-FeOOH). Other weight effects and slightly marked endothermic effects are visible at 490 °C. They result from the further breakdown (de-hydroxylation) of the structure of clay minerals—illite and kaolinite. At the temperature of 570 °C, the DDSC curve shows a well-marked endothermic effect; at the same time, it is not accompanied by any weight effect. It is related to the polymorphic transformation of low-temperature quartz into high-temperature ones. Above 600 °C, one very strong weight loss effect is visible on the DTG curve, with a maximum at 765 °C. It is accompanied by a clearly visible endothermic effect—the dissociation effect of calcium carbonate (calcite). At a similar temperature, a second endothermic de-hydroxylation of illite can also be observed. After the dissociation of the calcite is complete, no weight effects are observed, and the deviations of the DSC (heat flow curve—red one) and DDSC curve are the result of uneven heat flow between the sample and the thermocouple.

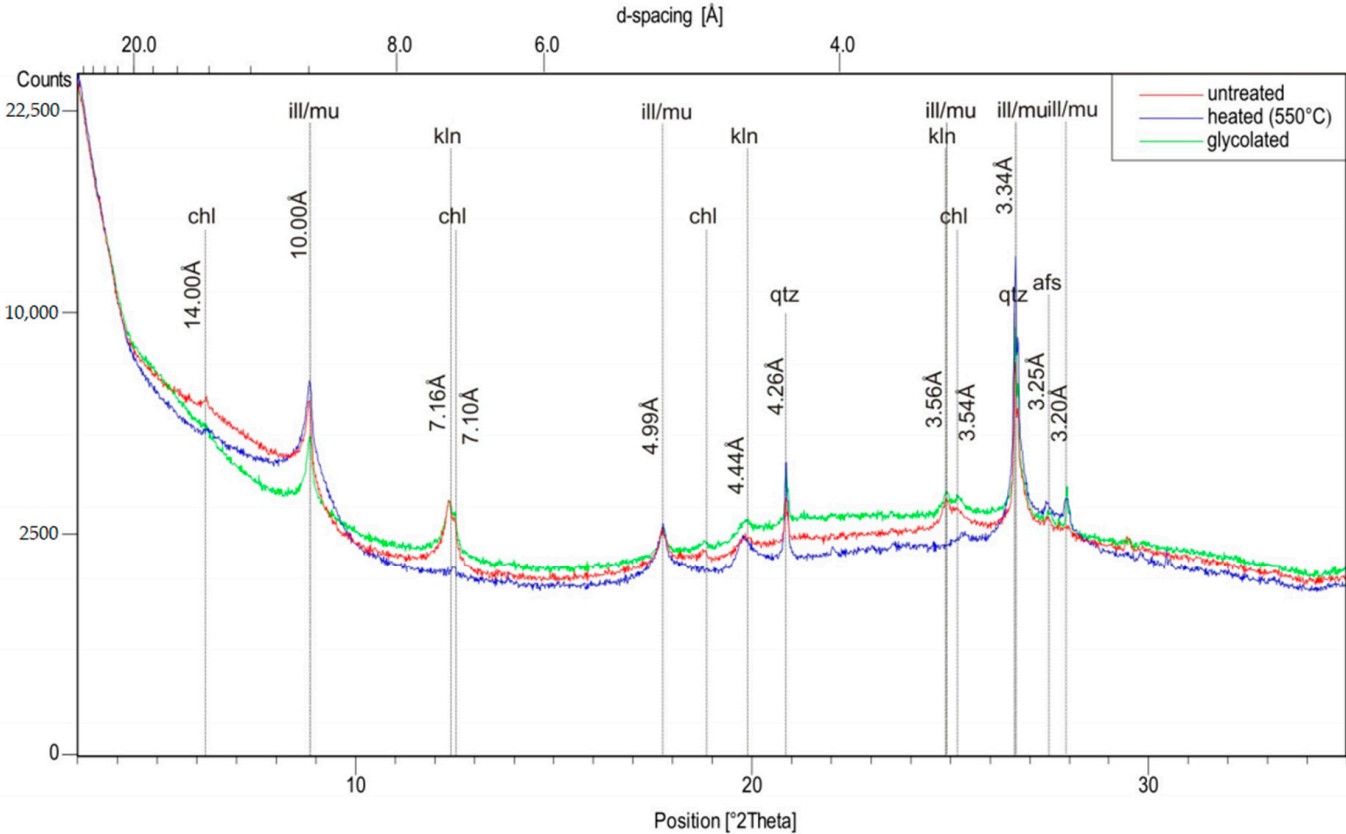

**Figure 7.** X-ray diffraction examination of clay fraction [40] (chl—chlorite, ill/mu—illite/muscovite, kln—kaolinite, qtz—quartz, afs—alkaline feldspar).

Based on sieve and aerometric analysis the amount of fractions in original mortar sample was: cl = 8%, si = 62%, sa = 30% (type of soil—sasiCl). On the basis of all studies, the substitutive material for laboratory tests carried out at the Faculty of Civil Engineering of the Warsaw University of Technology analysis was chosen.

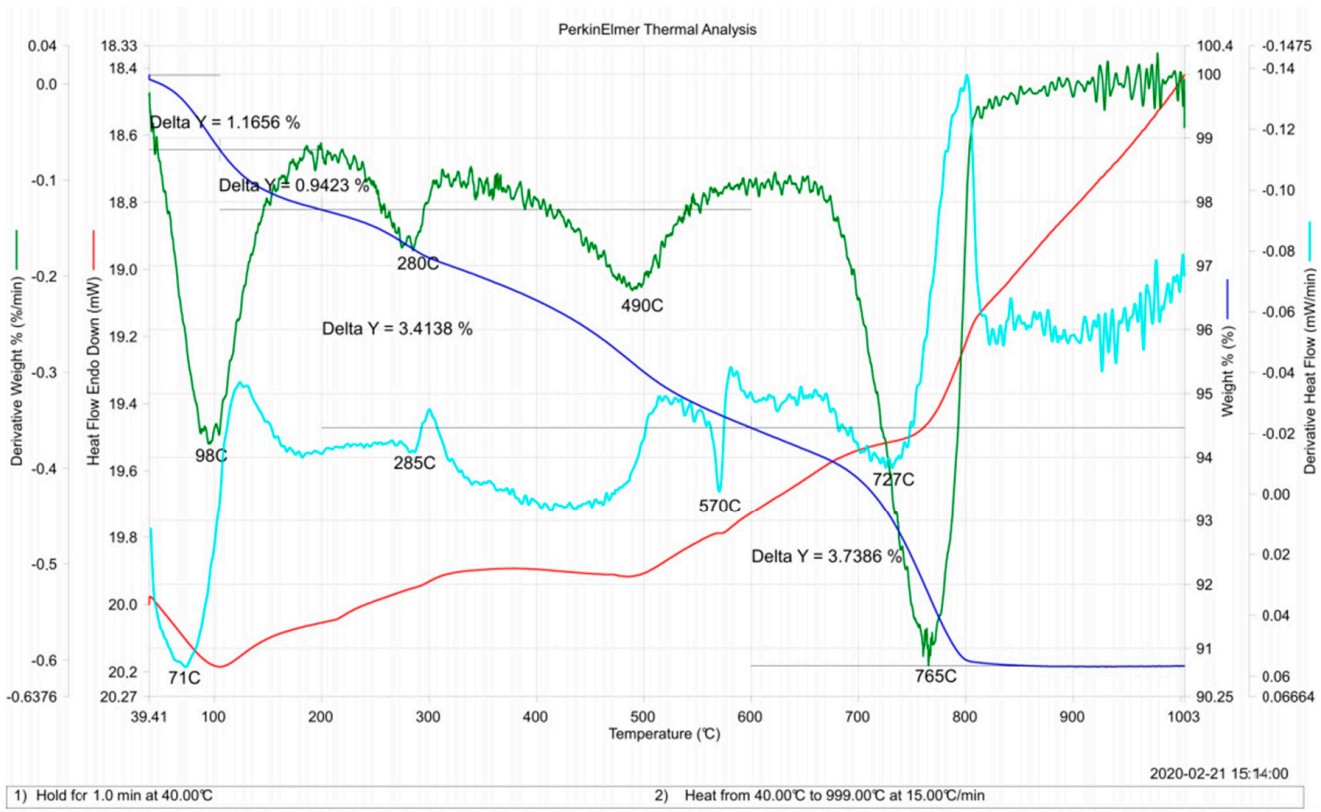

**Figure 8.** Thermogram of the sample [30] (the weight curve TG—navy blue; first derivative of the weight curve DTG—green; heat flow curve DSC—red; first derivative of heat flow curve DDSC—turquoise).

## 5. Properties of Earth-Based Mortar

The whole process to get the final composition of repair mortar, which can be used in the conservation works of stone masonry structures, is shown in Figure 9. As already mentioned, the earth is the main binder on the basis of which mortars used in irregular stone constructions in Tanais were made. Therefore, research to determine the properties of earth-based mortars used in masonry structures had to be carried out to determine the best way to strengthen it, using cement (creating an earth-based mortar stabilized with cement with the most appropriate recipe) and other substances available in regions where conservation works are carried out not only to improve the durability, physical and mechanical parameters but also to achieve the desired esthetic effect in the form of a suitable color (color combination in the wall) and compatibility of the repair mortar with the substrate. As a next step, the properties of earth-based mortar made on the base of substitutive material were tested. Since the exact proportions of ingredients in restored masonry walls were not specified, some test proportions were made for the research. Moreover, having in mind that tested mortar will be used in specific conditions of the archeological site, only volumetric dosing of ingredients can be used in this case.

The proportions of ingredients for the first test was 1:3 (earth:sand), and the amount of water used was 1:1 to the amount of earth. The workability of such composition is equal to 15.95 cm.

The results of the compressive strength test are presented in Table 1. Based on this, it can be confirmed that earth-based mortar belongs to a group of low-strength mortars, insufficient to be used in the original composition. According to the German standard DIN 18,946 (2018) [44], earth mortar for non-load-bearing walls must be sufficiently strong for the intended use, and usually, 1 N/mm$^2$ is sufficient. In polish withdrawn standard PN-B-14,501:1965 [45], mortar must have a strength of at least 0.8 N/mm$^2$. In both cases, the value, which we get is not enough. The freeze–thaw test showed that the earth-based

mortar sample, despite the use of the hydrophobizing agent, disintegrates already after placing it in a container with water for the first time (Figure 10), which proves its very low durability and resistance to weather conditions.

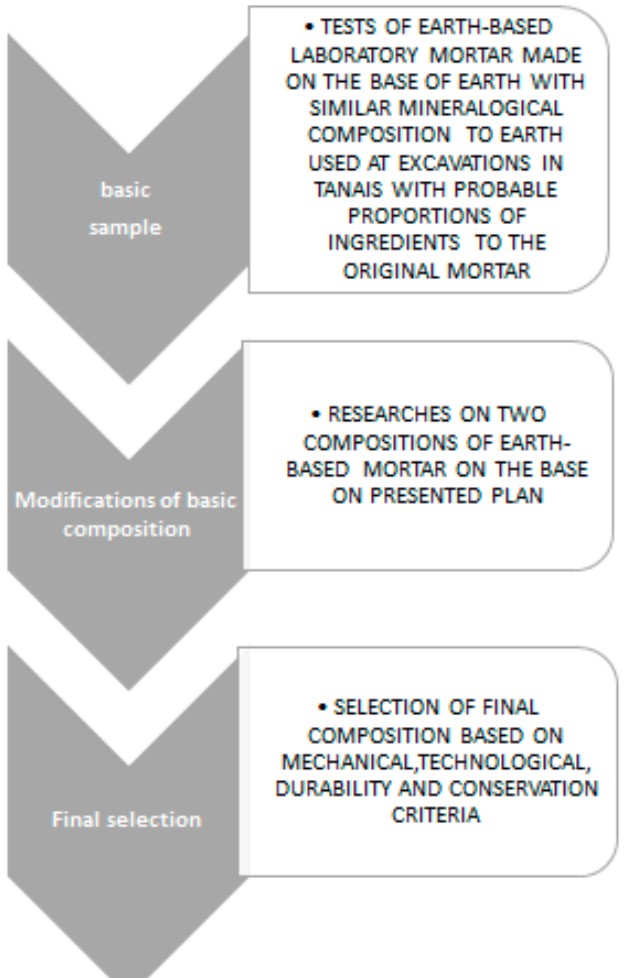

**Figure 9.** Steps to get the final composition of earth-based mortar for reconstructions of stone masonry walls.

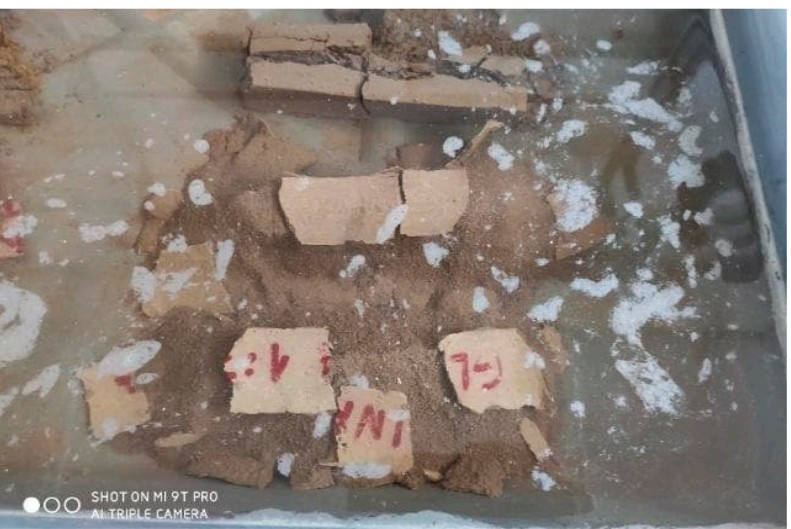

**Figure 10.** Clay mortar sample after the first cycle of freeze–thaw test.

**Table 1.** Compressive strength of earth-based mortar (sand:earth—1:3).

| X (MPa) | SD (MPa) | SD/c4 (MPa) |
|---------|----------|-------------|
| 0.61 | 0.09 | 0.10 |

X—arithmetic average; SD—standard deviation; SD/c4—unloaded estimator.

Earth-based mortar samples were also subjected to the test for determination of shrinkage and color analysis. In the case of mortar with sand: earth ratio of 3:1, the shrinkage was about 4.0%, which in practice led to cracks and the ingress of water into the wall, which largely contributed to the current state of the mortar. According to the new German standard [44], the maximum linear shrinkage should be equal to 2.5%. The color of the original mortar was assessed as RAL 1011 brown beige (Figure 11).

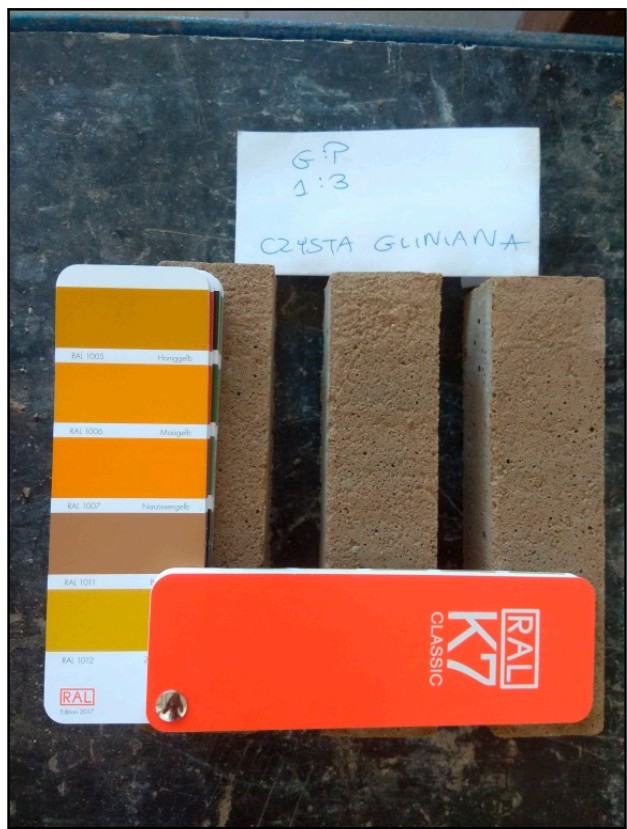

**Figure 11.** Color analysis of clay mortar sample using RAL K7 palette.

## 6. Modifications of the Composition of Earth-Based Mortar

Compatibility in the revitalization is a complex term, and in regard to the substrate it is applied to, is assessed in chemical (restoration mortar introducing hazardous compounds or compounds, which could interact in a negative manner with the substrate are forbidden), dimensional (e.g., modulus of elasticity, shrinkage), hygric adhesion (moisture transfer between the building materials in a homogenous manner) as well as historical and esthetical terms (The restoration mortar must have the same appearance as the authentic mortar) [46]. Taking into consideration, the idea to use mortar similar to the original one was the basis for modifications of composition. Since the insufficient properties of earth-based mortar, different types of stabilizations were taken into account [14,16–21]. Unfortunately, in the case of excavations in the Tanais, only cement and lime can be considered due to economical reasons as also accessibility of the materials in the conservation area. Based on literature studies [47] addition of lime will not provide desired mechanical properties and durability of the material, so the tests of the addition of concrete were performed. In the case of stabilization with concrete is especially important to observe and test possible soil

crystallization. Finally, based on standards on repair mortars for the historic masonry [1,22] as also on standards for earth-based mortars [44,48–51], the proposed mix design for test application as the first attempt of solving was chosen, and the whole test research program was created, which basic structure is shown in Figure 12. In the creation of the final composition, some main requirements were taken into consideration:

- Conservation issues:
  - mortar compatible with the existing masonry;
  - ease of removal of the mortar in future (mortar no stronger than needed for structural and durability reasons);
- Mechanical issues: withstand imposed permanent and transient loads.
- Durability issue:
  - Resist moisture ingress and airflow through the joint—appropriate finish of the mortar to encourage the shedding of water, low shrinkage, adequate bond with the masonry units, good workability, thereby increasing the likelihood of full contact;
  - Resistance to expected environmental loads (e.g., freeze–thaw);
- Technological issues—application of methods, equipment and materials available at the conservation area;

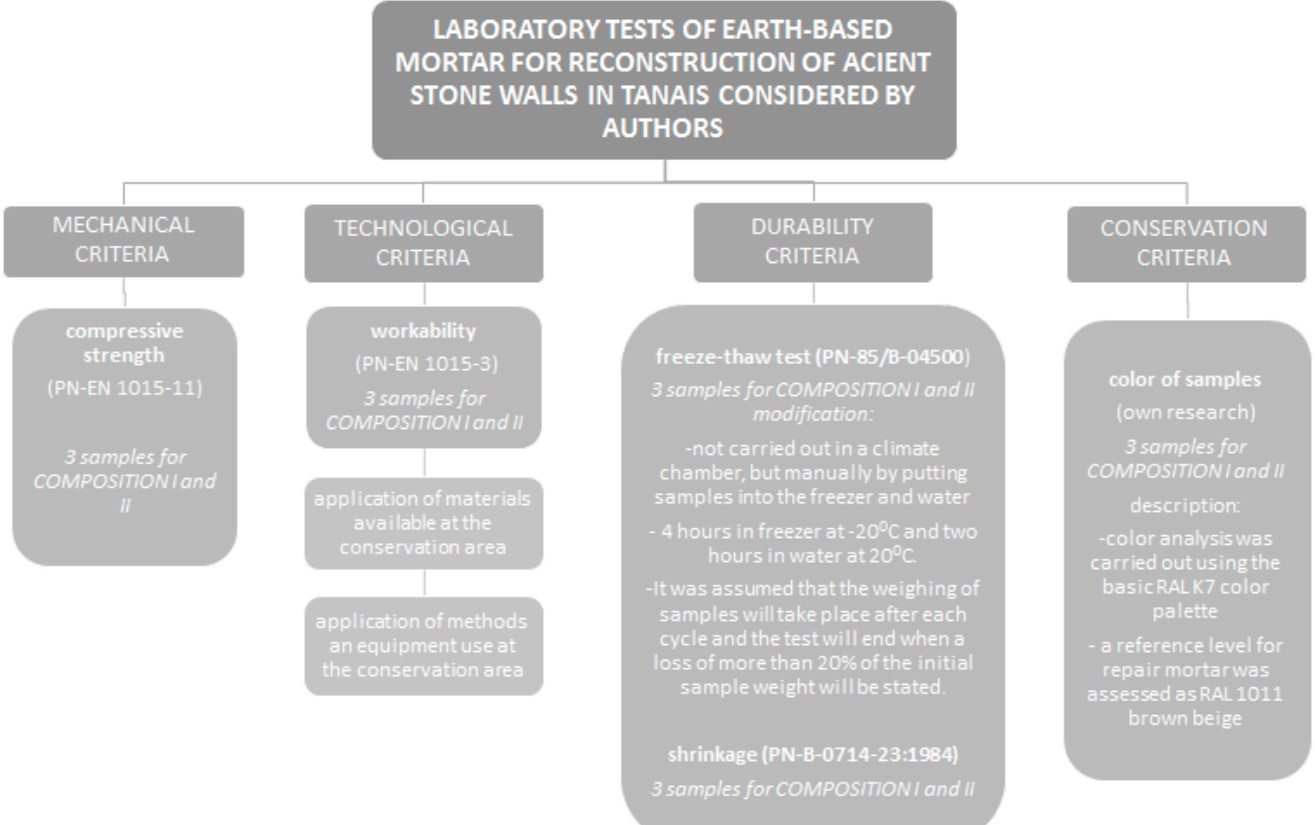

**Figure 12.** Plan of research for the final selection of clay mortar.

The tested compositions were:

- COMPOSITION I: earth-based mortar stabilized with cement, cement:earth:sand ratio 1:2:5;
- COMPOSITION II: earth-based mortar stabilized with cement, cement:earth:sand:lime 1:2:5:0.3;

In all of them, the amount of water used was 1:1 to the amount of earth and Portland cement CEM I was used. The compositions tested were based on a study of literature as the first idea of possible modifications. The research chosen covers the most important criteria for the mortar for the conservation, on the basis of which some additional parameters may be studied in the future. The first study was the workability analysis accordingly to PN-EN 1015–3 [41]. The results are shown in Table 2. Composition with the addition of lime has much better workability. The results of the compressive strength for compositions I and II are presented in Table 3. Comparing them with the compressive strength of earth-based mortar, it can be easily seen that the compressive strength of a cement-stabilized mortar is about four times bigger than the compressive strength of the original one. Additionally, the addition of lime has some favorable influence on the compressive strength.

**Table 2.** Workability of compositions I and II.

| No. | X (cm) |
|---|---|
| I | 20.5 |
| II | 18.2 |

**Table 3.** Compressive strength of compositions I and II.

| No. | X (MPa) | SD (MPa) | SD/c4 (MPa) |
|---|---|---|---|
| I | 2.55 | 0.13 | 0.14 |
| II | 3.55 | 0.15 | 0.17 |

X—arithmetic average; SD—standard deviation; SD/c4—unloaded estimator;

The next comparison was the frost resistance of new compositions. As already mentioned, earth-based samples not stabilized with cement were destroyed in the first cycle of the test. All new samples with the addition of cement survived eight freeze–thaw cycles. In all cases, the samples stabilized with cement were destroyed by the fracturing of the sample (Figure 13). It can be easily noticed that the surface layer of the sample is covered with a layer of a hydrophobizing agent. Due to the fact that also in the case of clay samples, this layer was intact, there is a presumption that water penetrated inside through leaks in the coating. It is recommended to perform these tests with an increase in the number of layers used, as well as to perform a durability test more suited to the actual situation of using the material. Comparing the shrinkage, the samples stabilized with cement showed smaller linear shrinkage than pure clay samples (about 3.0% for all compositions), but its value seems to be still too high for a repair mortar for stone structures exposed to changing atmospheric conditions. As was discussed in the introduction of this article, all materials used in the conservation activities should be original or compatible. In order to restore the original view of analyzed structures, the last comparison was color analysis. As in the case of clay mortar, the RAL K7 palette was used. The results of this test are presented in Figure 14. For all samples containing cement (compositions I and II), the color obtained is No. 7044 silk grey while for clay mortar No. 1011 brown–beige. It can therefore be concluded that the obtained color of mortar containing cement differs a little bit from the desired one, acquiring a grayish tone. Of course, the color of the earth will also have an influence on the final tone of composition, but it would be beneficial to limit the color change with the cement content or try to achieve the color in a different way.

Comparing all of the compositions tested, it can be stated that the modifications have a positive influence on the parameters of earth-based mortar. Finally, composition II was chosen for some test applications.

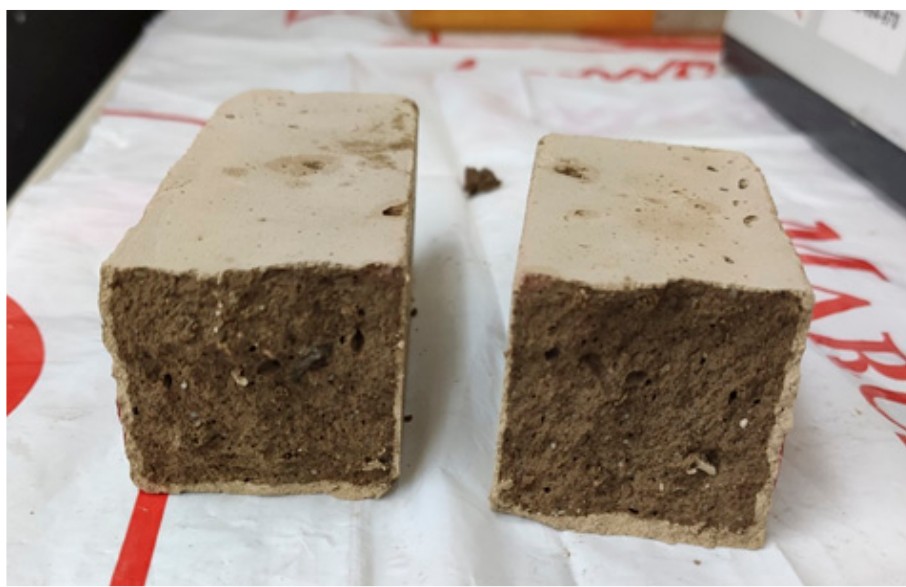

**Figure 13.** Composition I after eight freeze–thaw cycle.

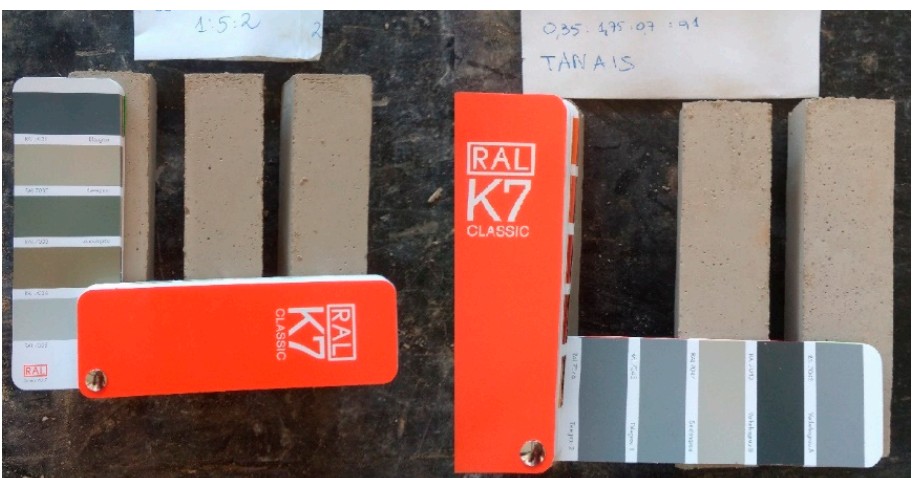

**Figure 14.** Color analysis of composition I and II using RAL K7 palette.

## 7. Application Analysis after Two Years

During the conservation actions at the Tanais site in 2017, the chosen earth-based mortar seem to fulfill the requirements both in the case of consistency and strength. In the next excavation season in 2018, the verification of applied treatment was also satisfactory since only the formation of small, superficial scratches and detachment of approximately 10 wall elements (stones) were observed. Damages and condition of restored masonry walls are shown in Figure 15. The resulting shrinkage cracks were secured and bonded by grouting with earth-based mortar stabilized with cement and plasticized with lime. During drying, the repaired joints were properly cared for to prevent further cracks, and the detached stones were rebuilt (after removing the previous mortar) and reintegrated with the wall. The view of the structure after repair works in 2018 is shown in Figure 16. After the restoration activities in 2017–2018, in 2019, no significant damage, superficial scratches or detachment of the wall elements (stones) were observed. There was also no visible destruction of the wall by atmospheric conditions and salt efflorescence on the face of the wall. Therefore, the properties of earth-based mortar in terms of mechanical, technological and durability issues may be assessed as satisfactory.

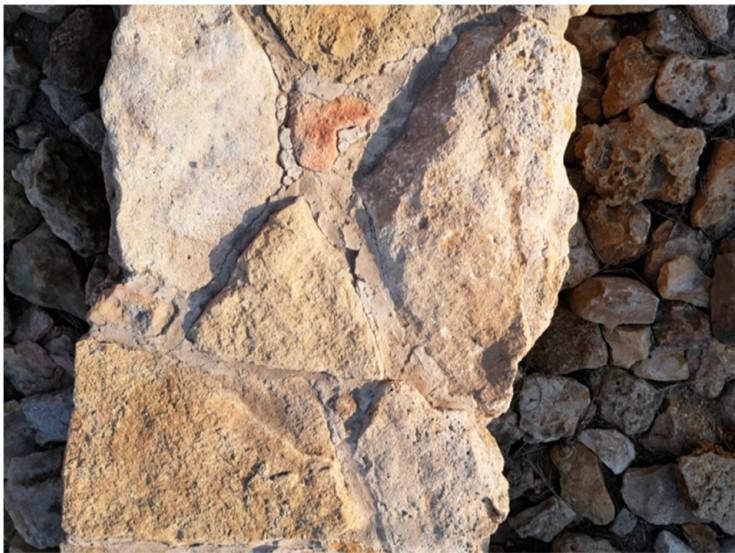

**Figure 15.** Cracks in earth-based mortar applied in 2017.

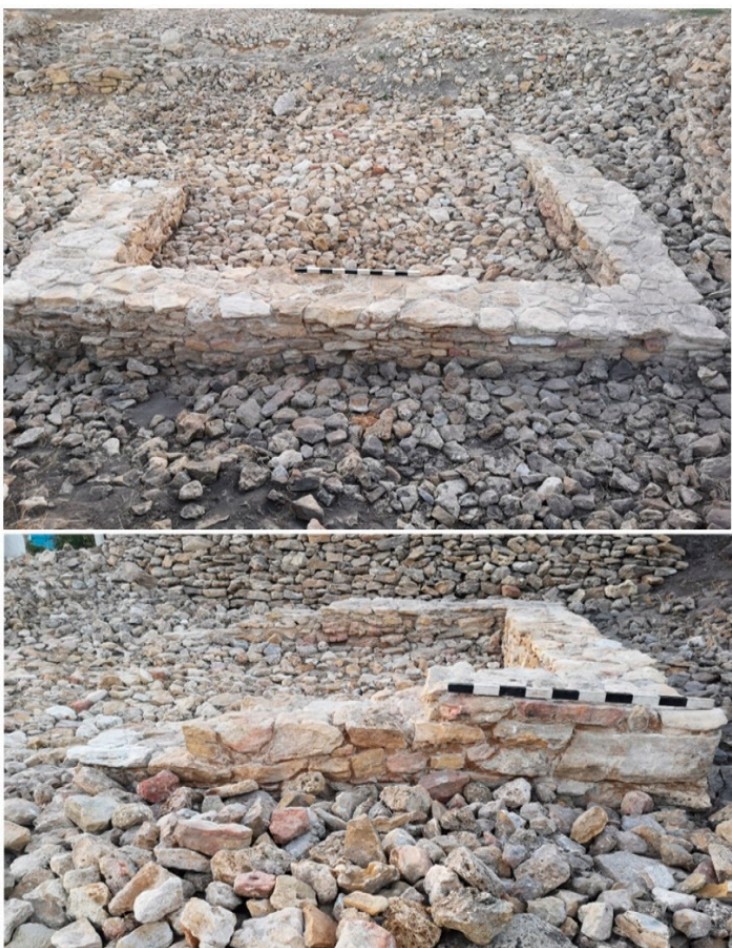

**Figure 16.** View of the wall after conservation, 2018.

The attempt to evaluate the applied treatment was also made using the methodology described in [24]. For each tested criteria, incompatibility risk was assessed using the scale rating from 0 (lack of incompatibility risk) to 10 (high incompatibility risk). All of the points are listed in Table 4. Incompatibility risk is quite small −1.2, but it may result from a moderate number of criteria taken into consideration. There is still a possibility to modify

the color of mortar so that it will be as similar to the original one as possible, but even this applied fulfill the conservation issues since it is not dominating over the whole structure as also to make a wider range of possible criteria in the future. In the next excavation season, some tests of salts development in the reconstructed wall are also planned to finally confirm the compatibility of applied repair mortar with the in-place stones.

**Table 4.** Evaluation of incompatibility index of applied test treatment.

| Criteria | Compatibility Indicators | Incompatibility Risks (Rating Scale) | IC |
|---|---|---|---|
| **CONSERVATION ISSUE** | | | |
| Visual properties | Colour difference | Little bit different than original but acceptable | → 5 |
| **MECHANICAL ISSUE** | | | |
| Mechanicial properties | Compressive strength | Values different but less than 10% | → 0 |
| **DURABILITY ISSUE** | | | |
| **Shrinkage** | Test | Little bit bigger than limit one but acceptable ofter test | → 5 |
| Frost resistant | Test | Smaller than limit one but acceptable after test | → 5 |
| **TECHNOLOGICAL ISSUE** | | | |
| **Workability** | Test | Good workability | → 0 |
| Availability of material | If the materials are available in conservation area | Yes | →0 |
| Costs | Cost comparing to other solutions | Acceptable | → 0 |
| IC – INCOMPATIBILITY INDEX | | $IC = \sqrt{\frac{R1^2 + R2^2 + \ldots + Rn}{n}}$ | 1,2 |

## 8. Conclusions and Planned Research

In the case of the archeological excavation in the Tanais applied solution seems to fulfill all the main requirements, so the test application can be evaluated as satisfactory. All the tests described above constitute the preliminary stage of research on the final selection of the best composition of the earth-based mortar, which could be used not only in archaeological sites in the Tanais but in all archeological sites where irregular masonry walls on earth-based mortars will be under conservation works. As it was presented, the problem of choosing the proper composition is a complex one that requires analyses of different criteria and ultimately selecting the best solution in a given case. In order to solve it, the expanded scope of research based on international standards for earth-based mortars [1,22,44,48–51] is planned. Apart from the research described in this article, some additional ones will be added, such as adhesion test, deformability and elasticity (Young's modulus), water protection (wet/dry test, vapor transmission). Some different compositions will also be examined—various amounts of cement and lime and eventually additions of an adhesion promoter (Torggler Neoplast latex modifier) and organic additives (eggs, fibers) [46], which may have positive influences on the color of mortar and its shrinkage [17]. Weights will be assigned to individual features, which can be modified depending on the encountered case of using an earth-based mortar, facilitating the appropriate selection of the best composition in a given case. The development of a full methodology for selecting the composition of an earth-based mortar for the preservation of ancient stone walls is an innovative issue, not yet developed, which makes it particularly valuable taking into account the large number of revitalized structures made on the basis of such material.

**Author Contributions:** Conceptualization, E.S. and M.G.-S.; methodology, E.S. and M.G.-S.; software, E.S. and M.G.-S.; validation, E.S., M.G.-S. and W.T.; formal analysis, E.S.; investigation, E.S.; resources, E.S. and M.G.-S.; data curation, E.S. and M.G.-S.; writing—original draft preparation, E.S.; writing—review and editing, E.S., M.G.-S. and W.T.; visualization, E.S. and M.G.-S.; supervision, W.T.; project administration, E.S.; funding acquisition, E.S. All authors have read and agreed to the published version of the manuscript.

**Funding:** This research received no external funding.

**Institutional Review Board Statement:** Not applicable.

**Informed Consent Statement:** Informed consent was obtained from all subjects involved in the study.

**Data Availability Statement:** The authors agree with Data Availability Statements.

**Conflicts of Interest:** The authors declare no conflict of interest.

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
