# Peer review of "Stability of Treatment from Earth-Based Mortar in Conservation of Stone Structures in Tanais, Russia"

_sustainability, doi:10.3390/su13042220_

Round 1

Reviewer 1 Report

Paragraph 2:   a short geographical and historical introduction to the Tanais site is necessary

line 101:         figure 4 is missing !

line 107.          if you speak about literature research, you should put the references !

lines 129-130 (Figure 5):        the observation in thin section at the optical microscope is a petrographical study, not a mineralogical study !

line 131:         aleurite is a small arborescent flowering plant in Euphorbiaceae genus

line 152:         the study of the clay minerals composition can be carried out only by x ray diffraction and, in case you have done it, you have to mention method and instrument

line 153:         you have to mention what you mean with aerometric analysis

lines 206-207: you have to mention which kind of cement you have used

Author Response

All points have been corrected.

Aleurite is also  is an unconsolidated sediment with a texture intermediately between sand and clay, similar to silt.

Reviewer 2 Report

The paper deals with a very interesting topic: the treatment of earth-based mortar in an archaeological site, but the international literature on restoration of mortars and plasters earth based is totally disregarded. Many references are in Polish.

The authors describe the application of restoration mortar cement based on original mortar earth based, neglecting any consideration on compositional compatibility.

The lack of aesthetic compatibility (different color of new mortars with respect the original ones) is evidenced by authors, but ignored.

The performed tests are conducted prevalently using PN (Polish regulations), difficult to examine for an international reviewer.

Furthermore any tests on salts development after the applications of new mortars on the archaeological site are performed.

Author Response

We tried to improve all the points mentioned

Reviewer 3 Report

general comment: the topic and analysis looks very appropriate to choose the best way to strengthen earth-based mortars. but there are some ambiguity in this research 1- have you used a valid research method used by other researchers that suggest these tests or this was based on your experience? 2- although the topic itself is valuable and worth reading, yet the idea to find a guideline for historic irregular structures in general (by focusing on one case study) could be more interesting for general readers. 3- you have to describe the findings of your previous researches related to this site and to historic irregular structures, and then show the development in this new work. 4- it would be way better to use graphic results for your analysis like graphs or charts - abstract needs a clarification; does this paper aim to describe this way? or introduce it? on the other hand it is better to mention the research method in the abstract. also a short note about findings would help readers to understand the whole concept of this paper by checking its abstract. in line 57; materials used in conservation activities should not necessarily have the same (or similar) mechanical and physical properties as 57 the original one. introduction: although it provides an appropriate background knowledge for entering this topic, it lacks clearness and the last paragraph is completely separated from other parts of introduction. and it seems that the last paragraph is the results of whole background, while its linkage to other parts is weak. it is also mentioned that historical masonry structures are often different is 3 categories, it is logical but do you have any reference or proof for this claim? section 2: in general it looks promising, how ever author should indicate the differentiation of this work with their previous work (Approach to conservation of irregular stone masonry based on archaeological excavations in the Black Sea basin; https://doi.org/10.1051/e3sconf/20184900117 in 2018) and if you use the same image in this paper you have to mention the reference. section 4: in Fig. 8 through which process you have chosen these criteria for final selection of clay mortar? is is based on "Metody badań zapraw do murów"??

Author Response

(The authors gave the same response as above.)

Round 2

Reviewer 2 Report

My suggestions were partially accepted and the paper has been insufficiently improved for the publication in Sustainability.

The paper is not well organized. 

Introduction :just  the international reference of RILEM was added, but no comparison was made with other international literature.

The paragraph 2 is too long and should be subdivided in sub-paragraphs.

The analytical section must be clearly separated from historical background.

A sub paragraph on environmental condition must be added.

Please, add a new paragraph with description of sampling, number of samples, images of samples , description of analytical methods...Just the XRD method  was described, petrographic and thermal methods are missing.

The spelling of some minerals is wrong ( i.e. illite  not ilite or illit, lepidocrocite, not lepidocroKite).

Please, improve the quality of Figure 4.

Paragraph 3 For me it is not possible to assess the correctness of the tests because international standards have not been used and I do not know Polish

The figure 7 is unreadable.

Figure 8: I prefer the use of a table. Please for decimal use dots, not comma.

The discussion must be improved with a comparison with international references.

Many of cited references are not internationally available. 

Author Response

-More literature in the introduction and discussion added

- paragraph 2 divided on sub-paragraphs, onesub-paragraphs for environmental condition, analytical part is a separate one

-other methods described, unfortunately we not have photos of sampels, only results

-spelling improved

-quality of figures improved

-paragraph 3 - some more detailed description of methods is provided

- figure 8 - another reviever asked to change one of tables on graph

Reviewer 3 Report

Authors have tried to improve the whole article and consider the reviewers suggestions. However, it has still some flaws including:

one of the major problem is that the international literature must be mentioned to indicate that this method is very crucial for the field, but the majority of references are in Polish and related to author of the paper and it might be because of the fact that this topic was not very interesting for the international researcher in the field.

the amount of self citation in this paper is around 20% which is not acceptable at all. (less than 10% is recommended)

grammatical errors (line 21 was/ lack of the article before part, ...)

_______________________________________________________________

  • Abstract:

the whole abstract is Ok, how ever it is better to see this concept as a way to reach the right composition for earth-based mortar. it means you have to show the gap in the field that the method to to finding the right compositions is something that researchers are really craving for. This point as a very important point of the paper should be more clear and constantly mentioned in the paper.

_______________________________________________________________

  • introduction; 

it is way better than the previous version. yet it has still some flaws, paragraph 2,3 are talking about the importance of identification of some aspects but they are separate. it is better to merge them and instead of exhaustive paragraph make it concise.

although you have deleted the last paragraph which seemed completely irrelevant to other parts of introduction, this section needs a paragraph to conclude.

_______________________________________________________________

  • Characteristics of irregular stone masonry in Tanais:

does Fig. 1 have  reference or not? has it published before?

_______________________________________________________________

  • Modifications of the composition of clay mortar:

this part now looks clear

_______________________________________________________________

  • Conclusion:

in conclusion the proposed process for reaching the right composition of earth-based mortar should be mentioned as a new and scientific method.

Author Response

  • some international literature added, self citation limited.
  • abstract and introduction impoved
  • fig 1 was not published before
  • conclusion improved

Round 3

Reviewer 3 Report

in this stage I can confirm that this paper is appropriate for being published.

Author Response

Some English changes was provided.